# Towards Continuous-time Causal Foundation Models

**Dennis Thumm** [1]  **Ruben Wiedemann** [2]  **Ying Chen** [3]

## Abstract

Previous causal Prior-Data Fitted Networks (PFNs) for time series are trained on discrete-time priors. A natural route to continuous time writes each mechanism as a stochastic differential equation (SDE), but integrating it once per observation gap makes the trajectory law depend on the observation schedule—leaving the prior discrete-time in all but name. We propose a precise criterion—a sampled trajectory's law must be invariant to the observation schedule—and a three-tier taxonomy (discrete; naive observation-grid integration; fine-grid integration decoupled from observation) that operationalises it. We realise the top tier with a prior over SDE-driven temporal SCMs on random DAGs, with Ornstein–Uhlenbeck or small-MLP drifts, irregular schedules, and hard / soft / time-varying interventions. A $2 \times 2$ encoder $\times$ integrator ablation finds that fine-grid integration beats naive across all seeds on a nonlinear prior—though the effect is null on a linear one—and that its lead grows as the evaluation grid is refined; a time-aware encoder helps only under naive integration. We release[1] the prior and a preliminary zero-shot protocol on pharmacokinetic and physical-system data.

## 1. Introduction

Prior-Data Fitted Networks (PFNs) (Müller et al., 2022; Hollmann et al., 2023; Nagler, 2023) pre-train a transformer on datasets sampled from an analytic data-generating prior and then perform in-context inference at test time. In causal settings, Do-PFN (Robertson et al., 2025) and CausalFM (Ma et al., 2026) have pushed this recipe to *inter-*

*ventional* tabular prediction by training on synthetic structural causal models (SCMs) (Pearl, 2009). Recent work extends causal PFNs to multivariate time series by sampling temporal SCMs (TSCMs) with lagged directed acyclic graphs (DAGs), nonlinear autoregressive mechanisms, and multiple intervention types (Thumm & Chen, 2026).

Existing temporal causal priors (Thumm & Chen, 2026) are *discrete-time*: the generating process steps on a regular integer grid and the lag structure is a stack of adjacency matrices indexed by integer offsets. A natural response is to rewrite the mechanism as a stochastic differential equation (SDE) and let it run between observations. But the devil is in the integration: if the SDE is stepped *once per observation gap* (Euler–Maruyama (EM) on the observation grid), the joint law of a trajectory depends on when it is observed, and the prior is still effectively a discrete-time Markov model in SDE clothing. The target domains that motivate continuous time—pharmacokinetic concentrations sampled at clinically chosen times (Boeckmann et al., 1994), physical systems like the Causal Chamber (Gamella et al., 2024) with variable-delay events, and electronic health records with missing-at-random and missing-not-at-random gaps (Che et al., 2018; Rubanova et al., 2019)—are *schedule-heterogeneous* and require more.

This paper takes a step back and asks what exactly a causal PFN prior must satisfy to be called continuous-time. Our contributions are:

1. A **precise criterion for continuous-time causal priors** (Section 3.1): the joint law of a sampled trajectory must be invariant to the observation schedule. We give a three-tier taxonomy—discrete ($\Delta t \equiv 1$), naive observation-grid integration, and fine-grid integration with decoupled observation—that operationalises the criterion.

2. A **construction that realises the top tier** (Section 3.2): Ornstein–Uhlenbeck (OU) or small-Multilayer Perceptron (MLP) nonlinear drifts on a random DAG with optional hidden confounders and Markov regime switches, irregular observation schedules, and hard / soft / time-varying interventions, all integrated on a fine grid and subsampled to the observation schedule.

3. An **empirical** $2 \times 2$ **encoder** $\times$ **integrator ablation** (Section 4) on a linear-OU and a nonlinear neural-drift

[1]National University of Singapore [2]Imperial College London [3]Department of Mathematics, Centre for Quantitative Finance, Risk Management Institute, National University of Singapore. Correspondence to: Dennis Thumm <dennis.thumm@u.nus.edu>.

*Proceedings of the $2^{nd}$ ICML Workshop on Foundation Models for Structured Data*, Seoul, South Korea. 2026. Copyright 2026 by the author(s).

[1]https://github.com/thummd/continuous-time-causal-pfn

prior. Fine-grid integration beats naive across all seeds on the nonlinear prior, with the lead growing as the eval grid refines—the discretisation-bias signature of Section 3.1; on the linear-OU prior the three-seed mean is null. The time-aware encoder helps only under naive integration, consistent with fine integration making the data-generating process approximately schedule-invariant.

Real-data transfer (Theophylline, Warfarin, Causal Chamber) is preliminary and deferred to Appendix D; the main body argues the continuity case on synthetic data where it can be measured cleanly.

## 2. Background and Related Work

**Causal PFNs.** Do-PFN (Robertson et al., 2025) and CausalFM (Ma et al., 2026) pre-train transformers on SCMs and estimate conditional interventional distributions in context on independent and identically distributed (i.i.d.) tabular data. They do not address temporal dependencies.

**Temporal interventional priors.** Only a handful of generators produce paired (observational, interventional) time-series data: CAnDOIT (Castri et al., 2024) restricts to hard interventions at known targets; TECDI/RealTCD (Li et al., 2023; 2024) handle soft or hard interventions in linear structural vector auto-regressive (SVAR) models; CaTSG (Xia et al., 2025) approximates do-calculus with a learned diffusion model. The most recent CausalTimePrior framework (Thumm & Chen, 2026) samples nonlinear autoregressive TSCMs with hard, soft, and time-varying interventions—but, like all of the above, on a discrete-time grid. We build directly on its lagged-DAG formulation (Boeken & Mooij, 2024) and replace the mechanism and schedule with continuous-time analogues.

**Continuous-time dynamical ML and SDE causality.** Neural ODEs (Chen et al., 2018), Neural SDEs (Kidger et al., 2021), and latent-ODE models for irregular series (Rubanova et al., 2019) demonstrate that continuous-time parameterisations can match or beat discrete ones on irregular data. Irregular-time attention (Shukla & Marlin, 2021; Tashiro et al., 2021) and time-series foundation models (Dooley et al., 2023; Taga et al., 2025; Moroshan et al., 2025; Xie et al., 2025) ingest continuous timestamps but, to our knowledge, none target *interventional* in-context prediction. Closest in spirit to our SDE-based prior, Lorch et al. (2024) *learn* a single SDE whose stationary distribution captures interventional behaviour, dropping acyclicity. Our goal is instead to *sample* an analytically specified prior over SDE-driven TSCMs so a transformer can amortise causal inference across the family; the two approaches are complementary.

## 3. Method

### 3.1. What makes a causal prior continuous-time?

Let $\mathcal{P}$ be a prior over (TSCM, trajectory) pairs, and let $\mathcal{P}_\tau$ denote the distribution of observations at schedule $\tau = (t_1 < \ldots < t_T)$.

**Definition 3.1** (Continuous-time causal prior). $\mathcal{P}$ is *continuous-time* if there exists a continuous-path stochastic process $X$ whose law is independent of $\tau$ and $\mathcal{P}_\tau$ is the law of $X|_\tau$. I.e. the observation schedule is pure measurement, not part of the TSCM.

The definition partitions priors into three tiers: (A) *discrete* ($\Delta t \equiv 1$), a VAR-style SCM (Thumm & Chen, 2026) defined only on the integer grid and failing Definition 3.1 by construction; (B) *naive continuous* (observation-grid integration), an SDE stepped once per observation gap $\Delta_i$—the joint kernel parameterises to a different Markov model as $\Delta_i$ varies, so the law depends on $\tau$; (C) *continuous* (fine-grid integration), the SDE integrated on $\Delta_{\text{fine}} \ll \min_i \Delta_i^{\text{obs}}$ and subsampled to $\tau$, converging to the true SDE law as $\Delta_{\text{fine}} \to 0$ (Kloeden & Platen, 1992). At any finite $\Delta_{\text{fine}}$ tier (C) realises Definition 3.1 only approximately, with $\|\mathcal{P}_\tau^{(C)} - \mathcal{P}_\tau^{(\text{SDE})}\| \to 0$ as $\Delta_{\text{fine}} \to 0$; we treat tier (C) as the practical realisation of the criterion.

Whether tiers (B) and (C) differ in practice depends on a stability condition. The standard Euler–Maruyama update on a 1-D OU process $dX = -\theta X \, dt + \sigma \, dW$ has amplification factor $|1 - \theta\Delta|$ per step and is mean-square stable only when $\theta\Delta < 2$; on a prior that crosses this boundary, naive EM produces exploding trajectories that pin training-target distributions at their normalisation ceiling—a numerical-stability artefact rather than a discretisation-bias signature. Stability is necessary but not sufficient for naive $\approx$ fine: the leading per-step Euler–Maruyama bias against the exact Gaussian OU transition kernel is $O(\theta\Delta)$ on the variance, accumulating over the trajectory at the prior's typical $\theta\Delta \approx 0.3$. Eval-loss is partially robust to this transition-kernel bias—it scores predictive likelihood, not path-measure distance—so we expect a smaller but still detectable empirical gap, which Section 4 confirms on both OU and neural priors. The construction (Section 3.2) therefore pairs tier-(C) integration with a stability-respecting prior, and the ablation tests both axes.

### 3.2. Construction of the continuous-time prior

**Graph sampling.** A sample from the prior draws $N \sim \text{Uniform}(3, N_{\max})$ variables and a DAG $\mathcal{G}$ over them (Thumm & Chen, 2026). We provide two graph samplers: (i) a named-structure sampler that cycles through eight canonical causal structures (back-door, front-door, instrumental variable, mediator, confounder-plus-mediator, observed confounder, unobserved confounder, bi-variate)

and (ii) a *random-DAG* sampler that draws an edge between each pair with a configurable probability $p$ under a topological ordering, with configurable probability that each non-$(A, Y)$ variable is marked *hidden* (removed from the encoder's input, but active in the dynamics). For each DAG we designate a treatment variable $A$ and an outcome variable $Y$ such that $A$ precedes $Y$ in topological order (see Appendix A).

**Mechanism family.** Unlike the per-lag adjacency stack used in discrete-time priors, the continuous-time prior reduces temporal dependence to a single parent set per variable. We support two drift families on that parent set. The *linear* drift is an Ornstein–Uhlenbeck mechanism (Øksendal, 2003)

$$dX_v = \Big(-\theta_v X_v + \sum_{u \in \mathrm{Pa}(v)} w_{vu} X_u\Big)dt + \sigma_v\, dW_v, \quad (1)$$

with $\theta_v > 0$, $\sigma_v > 0$, and $w_{vu} \sim \mathcal{N}(0, \sigma_w^2)$ sampled per TSCM. At $\Delta t \equiv 1$ this reduces to the AR(1) mechanism used by discrete-time causal priors (Thumm & Chen, 2026). OU admits an exact Gaussian transition kernel between any two times, so the linear-prior naive-vs-fine comparison should be read as EM-vs-EM rather than EM-vs-exact; we use EM uniformly across drift families because no closed form exists for the neural drift.

The *neural* drift replaces the linear parental sum with a small randomly-initialised two-layer $\tanh$-MLP $g_v$ on $\mathbf{z}_v = [X_v, X_{u_1}, \ldots, X_{u_k}]$:

$$dX_v = \big(-\theta_v X_v + s_v\, g_v(\mathbf{z}_v)\big)\, dt + \sigma_v\, dW_v, \quad (2)$$

with $g_v(\mathbf{z}) = \tanh\big(W_2 \tanh(W_1 \mathbf{z} + b_1) + b_2\big)$ and $s_v > 0$. We retain $-\theta_v X_v$ outside the MLP so trajectories stay bounded for any weight draw; the outer $\tanh$ bounds the nonlinear contribution to $[-s_v, s_v]$. Each trajectory draws the drift family per variable with a Bernoulli($p_{\mathrm{neural}}$) coin, so a single training run exposes the PFN to a mixture of linear and nonlinear dynamics.

**Regime switching.** Optionally (opt-in, off in the ablation), a fraction of trajectories is drawn from a *continuous-time regime-switching* TSCM—$R$ OU systems arbitrated by a sticky Markov transition matrix—letting the prior express structural breaks of the kind seen in pharmacology (absorption vs. elimination) and physical systems (see Appendix B).

**Observation schedule.** Given a horizon $H$ and an expected inter-observation gap $\bar{\Delta}$, we sample one of three schedules: *regular* ($t_i = i\bar{\Delta}$), *jittered* ($t_{i+1} - t_i = \bar{\Delta}(1+\xi_i)$) with $\xi_i \sim \mathrm{Uniform}[-\rho, \rho]$), or *Poisson* ($t_{i+1} - t_i \sim \mathrm{Exp}(1/\bar{\Delta})$). The model never sees the schedule as input; it only sees the actual timestamps.

**Simulation (fine-grid integration).** Rather than integrate once per observation gap, we pick a fine step $\Delta_{\mathrm{fine}} \ll \min_i \Delta_i^{\mathrm{obs}}$, integrate the SDE on the union grid via Euler–Maruyama (Kloeden & Platen, 1992) with per-step Brownian increments, and subsample at $\tau$. Setting $\Delta_{\mathrm{fine}} = \Delta_i^{\mathrm{obs}}$ recovers naive tier-(B); a regular unit-gap grid recovers tier-(A). The continuity ablation (Section 4) varies this single knob; the update rule is in Appendix B.

**Interventions.** For each sample we draw a target $i^\star$, a window $[t_{\mathrm{int}}^{\mathrm{start}}, t_{\mathrm{int}}^{\mathrm{end}})$ of duration between $10\%$ and $30\%$ of the horizon, and an intervention kind $\in \{\mathrm{hard}, \mathrm{soft}, \mathrm{time\text{-}varying}\}$:

$$\begin{aligned} \text{(hard)}\quad & X_{i^\star}(t) := c, \\ \text{(soft)}\quad & \mu_{i^\star}(X) \mapsto \mu_{i^\star}(X) + \delta, \\ \text{(time-varying)}\quad & X_{i^\star}(t) := c(t), \end{aligned}$$

active on the window. Hard-intervention values are optionally clipped to $[\mu_{i^\star} - 3\sigma_{i^\star}, \mu_{i^\star} + 3\sigma_{i^\star}]$ to keep the intervention inside the observed operating range of the target variable—analogous to the causal *positivity* (overlap) assumption (Hernán & Robins, 2020). The prior returns paired counterfactual and interventional trajectories by reusing the same Wiener noise across runs (cf. Pearl 2009, rung 3).

### 3.3. $\Delta t$-aware PFN encoder

We build upon a causal transformer encoder operating on a pre-intervention window (Thumm & Chen, 2026). Instead of a learned integer positional embedding we replace it with a Fourier embedding of continuous time:

$$\phi(t) = W_\phi\big[\sin(2\pi f_k t),\, \cos(2\pi f_k t)\big]_{k=1}^{K}, \quad (3)$$

with a geometric frequency bank $f_k \in [f_{\min}, f_{\max}]$ (defaults $0.01, 10$) and a learnable projection $W_\phi$. Times are referenced to intervention onset, $t \leftarrow t - t_{\mathrm{int}}^{\mathrm{start}}$, and inter-observation gaps $\Delta t_i$ are embedded with the same family after a $\log(1+\Delta t_i)$ transform to concentrate resolution at small gaps. The encoder is otherwise identical to the discrete baseline, enabling a controlled ablation.

At inference time we feed $(X_{\mathrm{obs}}, t_{\mathrm{obs}}, \mathrm{intervention\ spec}, t_{\mathrm{query}})$ and the model predicts the Gaussian (or quantile) distribution of $Y$ at $t_{\mathrm{query}}$ under the intervention.

### 3.4. Training

The prior runs on-the-fly during training; each batch draws a fresh TSCM, schedule, and intervention. We use either a quantile (pinball loss) or bar-distribution (Thumm & Chen, 2026) output head; full hyperparameters and architecture sizes in Appendix C.

*Table 1.* Encoder $\times$ integrator on `regular` eval ($s_{\text{eval}}{=}1$), both priors. Rows: trained integrator (*naive*: $s_{\text{train}}{=}1$, tier B; *fine*: $s_{\text{train}}{=}8$, tier C). Cols: encoder. Fine integration beats naive on every cell (4/4). Encoder gap (pos $-$ time, last column) is positive in every naive row and $\leq 0.0003$ in every fine row: with fine integration the encoder choice is empirically inert.

| Prior | Trained | pos | time | $\Delta_{\text{pos}-\text{time}}$ |
|---|---|---|---|---|
| OU | naive | 0.3714 | 0.3641 | +0.0073 |
|    | fine  | 0.3578 | 0.3575 | +0.0003 |
| Neural | naive | 0.3413 | 0.3405 | +0.0008 |
|        | fine  | 0.3354 | 0.3352 | +0.0002 |

*Table 2.* Integrator on `mixed` schedule (time-aware encoder) at two eval-grid refinements. Cols: trained integrator (*naive*: $s_{\text{train}}{=}1$; *fine*: $s_{\text{train}}{=}8$). Rows: eval-time substep tier $s_{\text{eval}}$ of the held-out test trajectories (independent of the model). Fine beats naive on every cell (4/4). *Fine's lead grows when the eval is more refined*: $+0.0018 \rightarrow +0.0057$ on OU, $+0.0048 \rightarrow +0.0088$ on Neural—an integrator-specific signature.

| Prior | $s_{\text{eval}}$ | naive | fine | $\Delta_{\text{naive}-\text{fine}}$ |
|---|---|---|---|---|
| OU | 1 | 0.3590 | 0.3572 | +0.0018 |
|    | 8 | 0.3826 | 0.3769 | +0.0057 |
| Neural | 1 | 0.3507 | 0.3459 | +0.0048 |
|        | 8 | 0.3608 | 0.3520 | +0.0088 |

## 4. Experiments

A $2 \times 2$ **encoder $\times$ integrator** ablation, run independently on a linear-OU and a nonlinear neural-drift prior (the latter is the $p_{\text{neural}}{=}0.5$ mixture that Appendix D calls *mixed*), separates the two axes (Tables 1, 2). The encoder axis is positional-only (learned absolute embedding, ablating the Fourier-time path) vs. time-aware (Section 3.3); the integrator axis is tier-(B) *naive* EM ($s_{\text{train}}{=}1$ substep per observation gap) vs. tier-(C) *fine* EM ($s_{\text{train}}{=}8$). Each prior trains four PFNs per seed (10 k steps; the tables report seed 0, with a three-seed replication in Findings), scored on held-out evals drawn from the same prior (all ablation TSCMs use the back-door topology; Appendix C).

**Eval distributions.** Two schedules crossed with eval-grid refinements $s_{\text{eval}} \in \{1, 8\}$ (best held-out eval-loss over 50 batches): `regular` (uniform $\Delta = 1$) and `mixed` (random per-trajectory regular / jittered / Poisson, the pretraining schedule). On `regular` the positional encoder's `arange(T)` positions equal the actual timestamps, so the two encoders see identical inputs at eval time—this isolates pos-vs-time gaps as training-side residue. The eval substep tier $s_{\text{eval}}$ probes the SDE limit independently of the model.

**Findings.** The single-seed tables show fine-grid integration winning on 8/8 fine-vs-naive comparisons, but three-seed replication finds this robust only on the neural prior

(4/4 cells, all seeds). On the linear-OU prior the effect is null: the three-seed mean is $\approx 0$ or slightly negative on half the cells (fine leads on 6/8 cells overall), and the single-seed OU win was a favourable draw. The single-seed sign-consistency ($p < 1/256$) does not survive replication; the two mechanistic signatures below do replicate across seeds. Crucially, fine's lead grows monotonically as the eval grid refines (Table 2, $+0.0018 \rightarrow +0.0057$ on OU and $+0.0048 \rightarrow +0.0088$ on Neural), which is the discretisation-bias signature predicted by Section 3.1: as the eval distribution approaches the SDE limit, the model trained at the SDE limit pulls ahead. The encoder axis is conditional on the integrator: robustly null with fine integration (the pos$-$time gap is within noise of zero across seeds) and only marginally time-aware-leading with naive. We read the interaction as follows: with fine integration the data-generating process is approximately schedule-invariant (Definition 3.1), so the model has little to gain from explicit time-gap features; with naive integration the conditional dynamics genuinely depend on $\Delta_i$, and the time-aware encoder's Fourier embedding of inter-observation gaps gives the model a route to compensate. The positional-only encoder is structurally OOD on `mixed` (positions $\neq$ times) and omitted from Table 2; its naive-vs-fine pattern mirrors time-aware. An instability check on $\theta_{\text{range}} = [0.5, 2.0]$ (where every naive-substeps batch saturated the target-normalisation clip) and PK / chamber zero-shot transfer are in Appendices D, E. Appendix F elaborates on the computational costs of the grid granularity.

## 5. Discussion and Limitations

**What the prior buys today.** A precise continuity criterion realised by tier-(C) integration. On the nonlinear prior, fine-grid integration robustly wins across seeds, including on the eval that matches the naive variant's training tier; on the linear-OU prior the effect is within seed noise. The encoder axis is conditional on the integrator: with fine, encoder choice is empirically inert; with naive it is not.

**Limitations and future work.** Per-cell $\Delta$s are small and the integrator effect is prior-dependent: robust on the neural prior, null on linear-OU. Within-regime noise is Markov; neural drifts capture nonlinear dependence but not time-correlated noise. Model capacity is small, real-data transfer (Section D) preliminary. Jump-diffusion SDEs and Neural-SDE drifts (Tzen & Raginsky, 2019) extend the construction; latent-ODE–style hidden states address non-Markov confounding.

## Impact Statement

This paper presents work whose goal is to advance the field of Machine Learning. There are many potential societal consequences of our work, none which we feel must be specifically highlighted here.

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

## A. Canonical TSCM structures

The named-structure sampler exposes the eight structures (back-door, front-door, instrumental variable, mediator, confounder-plus-mediator, observed confounder, unobserved confounder, bi-variate) in Figure 1. We reuse them as canonical sanity checks; the random-DAG sampler of Section 3.2 generalises this to any $N$ up to $N_{\max}$.

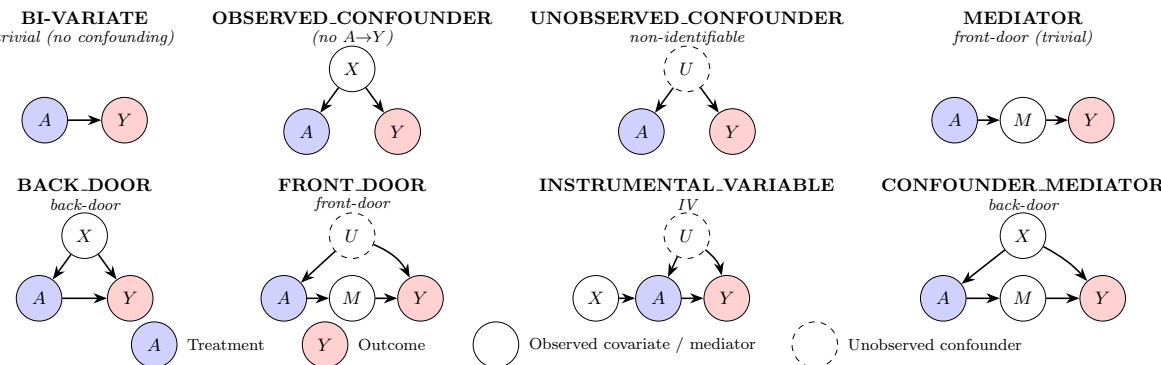

*Figure 1.* Canonical SCM structures used by the named-structure sampler. Each panel shows a back-door / front-door / IV-style template with the treatment $A$ (left), outcome $Y$ (right), and any mediators or confounders. The random-DAG sampler in Section 3.2 subsumes these as special cases. Figure adapted from Thumm & Chen (2026).

## B. Mechanism and simulation details

**Regime switching.**    A fraction of training trajectories is optionally drawn from a *continuous-time regime-switching* TSCM: $R$ independent OU systems that share variables and observation schedule, arbitrated by a sticky row-stochastic $R \times R$ Markov transition matrix ($P_{rr} \approx 0.9$, expected regime duration $\sim 10$ observations) with rows sampled from a Dirichlet distribution. This is off by default (trajectory fraction 0, Table 3) and not exercised in the ablation.

**Euler–Maruyama update and tier recovery.**    On the union grid $[t_1, t_T] \cap \{t_1 + k\Delta_{\text{fine}}\}_{k \geq 0}$ the integrator steps

$$X_v(t + \Delta_{\text{fine}}) = X_v(t) + \mu_v(X(t))\,\Delta_{\text{fine}} + \sigma_v\sqrt{\Delta_{\text{fine}}}\,Z, \quad Z \sim \mathcal{N}(0, 1),$$

with $\mu_v$ given by (1) or (2), and subsamples the result at $\tau$. Setting $\Delta_{\text{fine}} = \Delta_i^{\text{obs}}$ recovers naive tier-(B) integration; $\Delta_{\text{fine}} = 1$ on a regular unit-gap schedule recovers tier-(A). Brownian increments are re-sampled per fine step.

## C. Generator defaults and additional details

In Section 4 all other knobs are held fixed, with the following overrides of Table 3: tightened $\theta_{\text{range}} = [0.1, 0.5]$, which with the fixed schedule scale ($\bar{\Delta} = 1$, jitter $\rho = 0.3$) gives worst-case $\theta\Delta \leq 0.65 < 1$ on the regular and jittered schedules (Poisson gaps are unbounded, so the EM stability condition of Section 3.1 holds there only in probability—for $\theta = 0.5$ a single gap reaches the boundary $\theta\Delta \geq 2$ with probability $e^{-4} \approx 2\%$—but empirically $y_{\max}$ stayed below 5 across the grid, Section E); parental-weight scale $\sigma_w = 0.3$; back-door TSCM topology; *mixed* observation schedule (random per-trajectory choice between regular, jittered, and Poisson); 10 k training steps; batch size 24; $N_{\max} = 10$; embedding size 128, giving an identical model size across cells (1.1 M parameters).

## D. Preliminary zero-shot transfer to real irregular data

This appendix documents an early transfer study of the tier-(C) prior to three irregularly-sampled real-world datasets. We flag the numbers as corroborative and *not yet competitive* with dataset-specific baselines; treating them as a full zero-shot claim would require (i) broader mechanism families in the prior, (ii) calibration against domain baselines (NONMEM fits for PK, system-identification baselines for Causal Chamber), and (iii) sensitivity analysis to prior misspecification. We flag this as our primary line of future work.

*Table 3.* Generator default hyperparameters (`configs/continuous_default.yaml`). The experiments of Section 4 override several of these values; the overrides are listed in the text below.

| Group | Knob | Default |
|---|---|---|
| Graph | $N_{\max}$ | 16 |
| Graph | random-DAG edge prob. $p$ | 0.3 |
| Graph | hidden-conf. probability | $\{0.0, 0.3\}$ |
| Mechanism | $\theta_v$ | $\mathcal{U}[0.5, 1.0]$ |
| Mechanism | $\sigma_v$ | $\mathcal{U}[0.2, 0.6]$ |
| Mechanism | $w_{vu}$ | $\mathcal{N}(0, 0.5^2)$ |
| Mechanism | drift family (OU / MLP) | Bernoulli($p_{\text{neural}}$) |
| Mechanism | $p_{\text{neural}}$ | $\{0.0, 0.5\}$ |
| Mechanism | MLP hidden width | 8 |
| Mechanism | $s_v$ (MLP output gain) | $\mathcal{U}[0.5, 2.0]$ |
| Schedule | mean gap $\bar{\Delta}$ | 1.0 |
| Schedule | jitter $\rho$ | 0.3 |
| Intervention | window (frac. of horizon) | $\mathcal{U}[0.1, 0.3]$ |
| Intervention | kind probs. (H/S/TV) | $(0.5, 0.3, 0.2)$ |
| Regime-sw. | trajectory fraction | 0 (opt-in) |
| Regime-sw. | $R$ (regimes) | $\{2, 3\}$ |
| Regime-sw. | self-transition $P_{rr}$ | $\sim 0.9$ (Dirichlet) |
| Training | batch size | 32 |
| Training | total steps | 5,000 |
| Training | optimizer | AdamW ($\beta_1{=}0.9$, $\beta_2{=}0.98$) |
| Training | learning rate | 2e−4 |
| Training | LR schedule | cosine with warmup |
| Training | embedding size | 256 |
| Training | context window | 128 |
| Training | output head | quantile or bar |

**Theophylline pharmacokinetics.** The 12-subject NONMEM-distributed Theophylline dataset (Boeckmann et al., 1994): oral doses and 11 plasma-concentration measurements per subject over $\sim 24$ hours. We treat dose as a time-varying intervention and plasma concentration as the outcome $Y$. Times are converted to seconds and rescaled so that $\bar{\Delta}$ matches the training distribution; values are per-subject $z$-scored and rescaled back for reporting.

**Warfarin PK/PD.** 32 subjects with irregular oral-dose, plasma-concentration, and PD (prothrombin complex activity) observations (Hamberg et al., 2007). Variables are aligned to the canonical $(A, M, Y)$ front-door layout with dose as $A$, concentration as $M$, and PD as $Y$—the cleanest match to a named-structure TSCM sample from the prior. Per-variable $z$-scoring parallels Theophylline.

**Causal Chamber (wind-tunnel).** The light-tunnel `lt_walks_v1/actuators_white` benchmark used by earlier drafts produces uninformative causal-effect estimates (white-noise actuators, Pearson $r \approx 0$); see Section E for the failure analysis. We switch to `wt_intake_impulse_v1` (Gamella et al., 2024), the wind-tunnel impulse rig, which (i) carries an explicit binary `intervention` column—each $0{\to}1$ pulse marks a known toggle of the intake-fan setpoint `load_in` $\in \{0.01, 1.0\}$—and (ii) has real downstream dynamics on a 5-variable subgraph `load_in` $\to$ $\{\texttt{current\_in}, \texttt{rpm\_in}, \texttt{pressure\_intake}, \texttt{pressure\_downwind}\}$. We extract 200 episodes (50 pre / 20 post samples around each toggle, real per-row timestamps with median 0.15 s and max 2.4 s) and query each of three downstream variables; Pearson $r$ now varies meaningfully (Table 4).

**Eval protocol.** Each pretrained checkpoint is evaluated zero-shot. The PK adapter has the option to prepend $N$ synthetic pre-baseline observations (zero values, $z$-scored as $-\mu/\sigma$) so the encoder sees a non-empty pre-intervention window. We swept $N \in \{0, 2, 4, 8, 16\}$; $N{=}0$ **is best across both datasets and both mechanism families**, so the numbers reported in Table 4 use no padding. The full sweep (on the seed-0 checkpoint: mean Pearson $r$ on Warfarin cp stays in $[0.79, 0.89]$ across $N$ but drops elsewhere; on Theophylline mixed it flips sign from $+0.16$ at $N{=}0$ to $-0.54$ at $N{=}16$) confirms that the cross-variable mixer's empty-context fallback is acceptable as-is and that the augmentation *hurts* more often than it helps. We treat synthetic pre-baseline padding as a *negative result*: useful to know it doesn't earn its keep on these benchmarks, not as a method we recommend.

*Table 4.* Zero-shot transfer of the pretrained checkpoints (linear / mixed mechanism family, no eval-time padding $N{=}0$). **Warfarin and Causal-Chamber rows are mean $\pm$ std over five training seeds (0–4)**; Theophylline rows (†) remain single-seed. RMSE is the seed mean (std $< 2\%$ of the value); naive (constant-baseline RMSE) is seed-independent; lift and **Pearson** $r$ are mean $\pm$ std. **Pearson** $r$—the load-bearing dynamics-tracking metric—is highly seed-dependent: the previously reported single-seed headlines (Warfarin concentration $r \approx 0.88$, Chamber `rpm_in` mixed $r = 0.95$) were favourable draws that do not survive replication (see Findings). RMSE/lift are stable across seeds but, on the chamber, lift is regime-mean recovery rather than causal-effect tracking. Causal Chamber from 200 episodes of `wt_intake_impulse_v1/load_out_0.5_osr_downwind_4`.

| Dataset (variable) | Mech. | RMSE ↓ | naive | lift | Pearson $r$ ↑ |
|---|---|---|---|---|---|
| Theophylline (concentration)† | linear | 2.41 | 2.37 | $-1.8\%$ | $+0.53$ |
| Theophylline (concentration)† | mixed | 2.44 | 2.37 | $-3.2\%$ | $+0.16$ |
| Warfarin (concentration) | linear | 3.49 | 3.51 | $+0.5\pm1.1\%$ | $+0.49\pm0.69$ |
| Warfarin (concentration) | mixed | 3.53 | 3.51 | $-0.7\pm1.6\%$ | $-0.16\pm0.81$ |
| Warfarin (PD response) | linear | 25.21 | 25.25 | $+0.2\pm1.7\%$ | $-0.09\pm0.53$ |
| Warfarin (PD response) | mixed | 25.06 | 25.25 | $+0.8\pm2.5\%$ | $-0.05\pm0.57$ |
| Chamber-WT (`rpm_in`) | linear | 660.8 | 1264.8 | $+47.8\pm0.3\%$ | $+0.12\pm0.17$ |
| Chamber-WT (`rpm_in`) | mixed | 659.2 | 1264.8 | $+47.9\pm0.2\%$ | $+0.21\pm0.37$ |
| Chamber-WT (`current_in`) | linear | 77.1 | 159.8 | $+51.7\pm0.2\%$ | $+0.08\pm0.35$ |
| Chamber-WT (`current_in`) | mixed | 77.0 | 159.8 | $+51.8\pm0.1\%$ | $-0.16\pm0.01$ |
| Chamber-WT (`pressure_downwind`) | linear | 3.87 | 7.78 | $+50.2\pm0.3\%$ | $+0.01\pm0.05$ |
| Chamber-WT (`pressure_downwind`) | mixed | 3.87 | 7.78 | $+50.3\pm0.2\%$ | $+0.01\pm0.01$ |

**Findings.** Five-seed zero-shot transfer (Table 4) shows that the load-bearing Pearson $r$ is highly seed-dependent, while only the seed-stable but uninformative RMSE/lift quantities survive.

*(i) Warfarin plasma concentration:* Across five seeds the signal is unstable. Linear holds on four of five seeds ($r \in \{0.88, 0.86, 0.89, 0.72\}$) but the fifth flips to $-0.88$, giving $\bar{r} = +0.49 \pm 0.69$; mixed is sign-unstable seed to seed ($+0.89, -0.88, +0.76, -0.87, -0.73$), $\bar{r} = -0.16 \pm 0.81$. The PD outcome is centred near zero under replication ($\bar{r} = -0.09 \pm 0.53$ linear, $-0.05 \pm 0.57$ mixed). RMSE and lift are stable (RMSE std $< 2\%$) but uninformative, since both PK targets cluster narrowly around their per-subject means. The linear-mechanism PFN shows a positive concentration signal on most seeds but it is not reliable, and the mixed-mechanism PFN shows none. The qualitative ordering does survive—**the linear-mechanism PFN is more robust than the mixed under PK distribution shift**—but at a far weaker, higher-variance effect size.

*(ii) Wind-tunnel* `rpm_in`: The seed-0 mixed checkpoint attains $r = +0.95$ on `rpm_in`, which read as an unambiguous within-episode dynamics-tracking signal. The remaining four seeds give $r \approx \{0.10, 0.00, 0.00, 0.00\}$, so $\bar{r} = +0.21 \pm 0.37$; the linear cell is similarly weak ($\bar{r} = +0.12 \pm 0.17$). The faster sensors stay near zero on average (`current_in` $\bar{r} = -0.16 \pm 0.01$ mixed; `pressure_downwind` $\bar{r} \approx +0.01$). The $\sim 50\%$ RMSE lift is stable across seeds, but—as noted—that lift is regime-mean recovery, not causal-effect tracking; the metric that would distinguish the two (Pearson $r$) shows no reliable within-episode signal once seed variance is accounted for. The saturating-exponential ramp story may hold for the seed-0 model, but it is a property of one favourable initialisation, not of the prior or architecture. The PK-vs-chamber mechanism *flip* reported earlier likewise compared two single-seed outliers and does not survive replication. This sharpens our existing caveat that multi-seed replication is needed before any per-cell real-data claim can be trusted (Section 5).

**Schedule-invariant training does not rescue transfer.** The transfer checkpoints above are trained on a `regular` schedule with coarse integration ($s_{\text{train}}{=}2$), not the `mixed` schedule and fine integration ($s_{\text{train}}{=}8$) that the ablation (Section 4) advocates. Re-training both cells on that schedule-invariant config—matched architecture and prior, five seeds—does *not* improve real-data Pearson $r$: it stays $\approx 0$ and seed-unstable on Warfarin (cp $\bar{r} = +0.01 \pm 0.75$ linear, $+0.20 \pm 0.56$ mixed) and on the wind-tunnel chamber, where it is if anything slightly worse (`rpm_in` $\bar{r} = -0.39 \pm 0.47$ linear, $-0.20 \pm 0.40$ mixed, versus $+0.12$ / $+0.21$ under the original config), with RMSE unchanged. The weak transfer therefore reflects a prior-to-real-data gap rather than the training schedule or integration grid.

**Domain-specific baselines.** Table 5 compares the CT-PFN against system-identification baselines on `rpm_in` (identical wt rig, subgraph, and 50/20 episode protocol; baselines fit on the chamber data, the PFN zero-shot). First, the constant-baseline RMSE in Table 4 (persistence: last pre-intervention value held forward, 1265) is a *weak* baseline; a post-regime-mean predictor (`Zero`, 654) already matches the PFN ($659 \pm 2$), so most of the "$+48\%$ lift over naive" reflects beating persistence,

not causal tracking. Second, the strongest baseline—vector autoregression conditioned on the intervention (`VAR_Int`, 645)—*edges the PFN on RMSE*. On PK the gap is starker: a textbook 1-compartment oral model $C(t) = A(e^{-k_e t} - e^{-k_a t})$ fit per subject attains pooled RMSE $0.62$ vs. the PFN's $2.41$ on Theophylline and $0.87$ vs. $3.49$ on Warfarin concentration, with Pearson $r \approx 0.97$–$0.98$ against the PFN's $0.49$–$0.53$. This individual fit is in-sample and hence optimistic, but it is also the standard pharmacometric workflow, and the $\sim 4\times$ RMSE gap leaves no doubt that the PFN is not competitive on accuracy. The CT-PFN's distinguishing property is zero-shot transfer with no per-dataset fit, not accuracy supremacy.

*Table 5.* CT-PFN vs. system-identification baselines on Causal-Chamber-WT `rpm_in` (pooled RMSE ↓; baselines from the sibling discrete-time pipeline, identical protocol). The PFN is zero-shot; baselines are fit on the chamber data. `VAR_Int` (vector-AR conditioned on the intervention) is the strongest and edges the PFN; a post-mean predictor (`Zero`) already matches it.

| Method | pooled RMSE ↓ |
|---|---|
| Persistence (naive, Table 4) | 1265 |
| AR1 | 1252 |
| VAR_Obs | 1226 |
| VAR_CE | 671 |
| Zero (post-mean) | 654 |
| CT-PFN (ours, 5-seed) | $659 \pm 2$ |
| **VAR_Int** (VAR + intervention) | **645** |

# E. Additional failure modes and caveats

Three concrete failure modes surfaced during the development of this prior; all three reshaped the experimental design in ways worth documenting.

**Clip-saturation pathology under unstable priors.** An early grid trained on $\theta_{\mathrm{range}} = [0.5, 2.0]$ ($\theta\Delta$ up to 2.0 on the regular and 2.6 on the jittered schedule—at and above the EM stability boundary $\theta\Delta < 2$, so the Euler self-loop factor $1 - \theta\Delta$ oscillates in sign step to step) had every naive-substeps batch saturate the $\pm 10\,\sigma$ target normalisation clip on at least one sample. Roughly half the fine-substeps batches did the same, through a second route: with the original parental-weight scale $\sigma_w = 0.5$, the drift matrix's spectral radius was comparable to $\theta$, so multi-variable coupling exploded even under fine integration. The resulting "naive-vs-fine" gap there was a numerical-stability artefact rather than a discretisation-bias signature. Tightening the prior to $\theta_{\mathrm{range}} = [0.1, 0.5]$, lowering $\sigma_w$ to 0.3, and raising the clip ceiling to $\pm 50$ pushed empirical $y_{\max}$ below 5 across the entire grid; with the artefact removed, the residual (B)-vs-(C) integrator gap reported in Section 4 is what the discretisation-bias accounting of Section 3.1 predicts. This is the empirical motivation for the stability condition.

**Zero-context-augmentation broke per-variable normalisation.** A separate training-time fix for the PK regime (where the encoder sees an empty pre-intervention window) used to fire a Bernoulli($p_{\mathrm{no\_context}}$) coin per sample to force `int_onset_idx = 0`. The downstream per-variable $z$-scoring then computed mean and std over an empty pre-window—the masked statistics fell back to $(\mu, \sigma) = (0, \epsilon)$ with $\epsilon = 10^{-2}$, blowing $Y_{\mathrm{true,norm}}$ up by a factor of $\sim 100$ and pinning the targets at the new clip. Eval loss climbed monotonically with $p_{\mathrm{no\_context}}$ ($0.34 \to 1.1 \to 2.3$), the opposite of what the augmentation was meant to achieve. The eval-side counterpart (synthetic pre-baseline padding in the PK adapter, Appendix D) avoids the issue because the prepended zero rows make the pre-window non-empty.

**The `actuators_white` chamber benchmark motivated a benchmark switch.** Earlier drafts evaluated chamber transfer on the light-tunnel `lt_walks_v1` `actuators_white` experiment, with episodes defined by a change-point detector on the eight polarizer / lamp actuator columns. That benchmark turned out to be *structurally* unsuited to a causal-effect claim. Every actuator (`pol_1`, `pol_2`, `l_11`, ...) is independently white-noise-driven: $> 99\,\%$ of consecutive samples have step changes $> 0.5$ in every actuator simultaneously. The "intervention episodes" the detector finds are not interventions in the SCM sense; they are cross-sections of a continuously-randomised process. The post-intervention variance of the queried sensor (`red`) is $95\,\%$ within-episode dynamics and only $5\,\%$ between-episode regime mean, and Pearson $r$ between any model's predictions and ground truth is statistically zero. Apparent "lift over naive" on this dataset is regime-mean recovery, not causal-effect tracking. The other two `lt_walks_v1` experiments (`smooth_polarizers`, `color_mix`) have continuous actuator sweeps and produce zero episodes under any reasonable change-point heuristic. The wind-tunnel `wt_intake_impulse_v1` dataset, used in Table 4, fixes all three structural issues at once: explicit binary

`intervention` column (no change-point heuristic), real physical-system dynamics (`rpm_in` ramps over $\sim 20$ samples), and real per-row timestamps with non-trivial jitter. The wt rig is therefore a sounder benchmark than the lt rig; note, however, that the eye-catching seed-0 `rpm_in` value of $r = +0.95$ does *not* survive five-seed replication ($\bar{r} = +0.21 \pm 0.37$; see Findings), so the wt rig fixes the benchmark's structure without yet yielding a reliable positive transfer result.

## F. Computational cost of grid granularity

The integration substep count $s_{\text{train}}$ enters *only* the synthetic prior's Euler–Maruyama data generator, as $(T-1) \cdot s_{\text{train}}$ integration steps (Section 3.2); it leaves the model architecture, parameter count, activation memory, and per-step compute untouched. Two consequences (Table 6, measured on one A100, batch 32): (i) **inference and deployment cost are invariant to the training grid**—a model trained at $s_{\text{train}}=8$ runs at identical cost to one trained at $s_{\text{train}}=1$, so finer grids impose no overhead in resource-constrained or large-scale deployment; (ii) the cost is paid once, at training time, in data generation, and scales *linearly* ($s_{\text{train}}=8$ is $7.4\times$ costlier per batch than $s_{\text{train}}=1$; at $s_{\text{train}}=8$ data generation dominates the step, 8.7 s vs. 32 ms of model compute). Since the naive→fine accuracy gain saturates by $s_{\text{train}}=8$ (Tables 1–2: eval-loss $\Delta \leq 0.015$, and within seed noise on the linear-OU prior), we recommend $s_{\text{train}}=8$ as a conservative default and note $s_{\text{train}}=2$–4 retains most of the benefit at a quarter to half the data-generation cost. The current generator is an un-vectorised Euler–Maruyama loop; vectorising across sub-steps would remove most of this training-time overhead.

*Table 6.* Cost of integration granularity (one A100, batch 32). $s_{\text{train}}$ enters only the prior's data generation; data-generation time scales linearly with $s_{\text{train}}$, while model forward+backward time and GPU memory are invariant. Inference/deployment cost is independent of the training grid.

| $s_{\text{train}}$ | data-gen (ms/batch) | $\times$ vs. $s=1$ | model fwd+bwd |
|---|---|---|---|
| 1 | 1177 | $1.0\times$ | |
| 2 | 2254 | $1.9\times$ | |
| 4 | 4377 | $3.7\times$ | 32 ms (flat) |
| 8 | 8667 | $7.4\times$ | |
| 16 | 17199 | $14.6\times$ | |
| 32 | 34361 | $29.2\times$ | |

