# OpenReview forum: "Towards Continuous-time Causal Foundation Models"
_ICML.cc/2026/Workshop/FMSD — FMSD @ ICML 2026 Poster_

### Official Review · Reviewer_n6sc · 2026-05-13
**Good motivation but lacking granularity ablations**

**Rating:** 6
**Confidence:** 4

**Review:**

This paper presents a compelling framework for handling irregular time series in causal inference by modeling underlying mechanisms with continuous stochastic differential equations (SDEs). By utilizing a "fine-grid integration" strategy that cleanly decouples the true system dynamics from sparse, irregular observation schedules, the authors demonstrate statistically significant performance gains over naive, observation-dependent integration methods. However, a critical shortcoming of this work is the conspicuous absence of any discussion regarding the trade-off between grid granularity and computational cost. While the experiments show that an arbitrary fine-grid setting (e.g., 8 substeps) outperforms a 1-step baseline, the paper completely omits an analysis of how increasing this resolution impacts training time, memory consumption, or the point of diminishing returns in accuracy. Without a rigorous cost-benefit analysis detailing how the computational overhead scales with finer integration steps, the practical viability of deploying this otherwise promising continuous-time architecture in resource-constrained or large-scale real-world scenarios remains an open question.

---

### Official Review · Reviewer_wQCz · 2026-05-20
**Towards Continuous-time Causal Foundation Models**

**Rating:** 7
**Confidence:** 3

**Review:**

The authors propose transitioning causal PFNs from discrete to continuous time by establishing a criterion for trajectory law invariance to observation schedules. Introduced a three tier integration taxonomy and construct a prior with linear and non-linear drifts on random DAGs, utilizing fine grid integration. Ablation studies confirm that fine grid integration outperforms naive observation grid integration, reducing the need for explicit time gap encoding.  Zero shot tests on real world datasets show promise in capturing system dynamics.

Strengths
1. The proposed data-generating process robustly handles linear/non-linear drifts, random DAGs, and multiple intervention types.
2. The 2×2 ablation isolates and validates the benefits of fine-grid integration over naive approaches pretty well.

Areas for Improvement
1.  Should do a comparison against strong domain specific baselines.